# Detection of *Candida albicans*-Specific CD4+ and CD8+ T Cells in the Blood and Nasal Mucosa of Patients with Chronic Rhinosinusitis

**DOI:** 10.3390/jof7060403

**Published:** 2021-05-21

**Authors:** Pascal Ickrath, Lisa Sprügel, Niklas Beyersdorf, Agmal Scherzad, Rudolf Hagen, Stephan Hackenberg

**Affiliations:** 1Department of Oto-Rhino-Laryngology, Plastic, Aesthetic and Reconstructive Head and Neck Surgery, University of Wuerzburg, 97080 Wuerzburg, Germany; lisa.spruegel@gmx.de (L.S.); scherzad_a@ukw.de (A.S.); hagen_s@ukw.de (R.H.); hackenberg_s@ukw.de (S.H.); 2Institute for Virology and Immunobiology, University of Wuerzburg, 97078 Wuerzburg, Germany; niklas.beyersdorf@vim.uni-wuerzburg.de

**Keywords:** *Candida albicans*, chronic rhinosinusitis, T cell activation, nasal polyps

## Abstract

*Candida albicans* is ubiquitously present, and colonization in the nose and oral cavity is common. In healthy patients, it usually does not act as a pathogen, but in some cases can cause diseases. The influence of *C. albicans* as a trigger of T cell activation on the pathogenesis of chronic rhinosinusitis (CRS) is controversial, and its exact role is not clear to date. The aim of the present study was to detect and characterize *C. albicans*-specific CD4+ and CD8+ T cells in patients with CRS, with and without nasal polyps. Tissue and blood samples were collected from patients suffering from chronic rhinosinusitis with (CRSwNP) and without nasal polyps (CRSsNP), and from healthy controls. A peptide pool derived from *C. albicans* antigen was added to tissue and blood samples. After 6 days, lymphocytes were analyzed by multicolor flow cytometry. Activation was assessed by the intracellular marker Ki-67, and the cytokine secretion was measured. Tissue CD8+ T cells of CRSsNP patients showed a significantly higher proportion of Ki-67+ cells after activation with *C. albicans* antigen compared to peripheral blood CD8+ T cells. Cytokine secretion in response to *C. albicans* antigen was similar for all study groups. In this study, *C. albicans*-specific CD4+ and CD8+ T cells were detected in peripheral blood and mucosal tissue in all study groups. In patients suffering from CRSsNP, *C. albicans*-specific CD8+ T cells were relatively enriched in the nasal mucosa, suggesting that they might play a role in the pathogenesis of CRSsNP.

## 1. Introduction

Chronic rhinosinusitis (CRS) is subclassified into two groups of disease according to its phenotypical appearance: chronic rhinosinusitis with nasal polyps (CRSwNP), and without nasal polyps (CRSsNP) [1]. CRS is defined as a chronic disease when symptoms last at least 12 weeks without complete disappearance [1]. Aside from this historical subdivision, more recent classifications focus on the different types of inflammation. Tomassen et al. introduced a categorization into different endotypes [2]. CRSwNP is described as a mostly T helper (Th) 2 inflammation, while CRSsNP is dominated by a Th1/Th17 signature [2]. Different settings of T cells and their cytokines have been previously described in the literature [3,4].

*Candida albicans* belongs to the yeast fungus family, and is considered to be an opportunistic pathogen [5]. *C. albicans* digestion by macrophages requires fungus internalization in phagosomes, followed by macrophage activation by Th cells. This activation process is mainly mediated by Th1 and Th17 cells [6,7]. After passing the epithelial barrier, *C. albicans* is identified by the immune system and the inflammatory response is initiated [8]. However, *C. albicans* is able to bypass the innate immune system, which plays a role in invasive candidiasis in patients with immune disorders [9]. *C. albicans* is one of the most commonly found fungal pathogens in the mucosa of the upper human aerodigestive tract. Both CD4+ and CD8+ T cells are involved in the body’s defense against a fungal infection caused by *C. albicans* [10].

Especially in CRSwNP, bacterial [11] and fungal [12] triggers can directly influence the inflammatory response. Activation of T cells by *Staphylococcus aureus* superantigen [13,14] has already been identified as contributing to the pathogenesis of CRSwNP, and was introduced into the classification of endotypes [2]. To date, the role of fungi in the pathophysiology of CRS remains unclear, but previous studies have shown differences in inflammatory markers in patients with or without fungal colonization [15,16,17]. Nasal colonization with fungi is frequent, and even more frequent in patients with CRS than in healthy controls [18]. Ponikau et al. found that 21.4% of CRS patients were colonized with *Candida* in comparison to 7.1% of healthy patients [18]. In patients with CRSwNP, an increased secretion of interleukin (IL)-6 or IL-8, mediated by exposure to extracts of *Candida parapsilosis* and *Rhodotorula mucilaginosa*, was observed [12]. Furthermore, CRS is associated with defects in the epithelial barrier, such as the downregulation of tight junctions [1]. This barrier dysfunction may lead to an increased interaction between fungi and the innate immune system, with activation of T cells and higher cytokine levels.

To our knowledge, no study has focused on the activation effects of antigens from *C. albicans* on local tissue CD4+ and CD8+ T cells when comparing patients with different types of CRS and healthy controls. The aim of this study was to measure CD4+ and CD8+ T cells activated by *C. albicans* antigens in CRS patients in vitro, as a possible pathomechanism of CRS.

## 2. Materials and Methods

### 2.1. Ethical Issues

The study was approved by the local ethics board of the Wuerzburg University Medical School (No. 116/17), and all participants gave written informed consent.

### 2.2. Isolation of Human Lymphocytes

Isolation of human lymphocytes was performed as previously described [3,19]. Intraoperative blood samples were collected and taken to the laboratory. Separation of the lymphocytes was carried out via density gradient centrifugation (10 min, 1000× *g*) at room temperature (RT), with equal amounts of Ficoll (Biochrom, Berlin, Germany) and equal amounts of peripheral blood. In order to determine the cells’ number and viability, a cell counter and analyzer system (CASY TT, Innovatis, Reutlingen, Germany) was used according to the manufacturer’s specifications. After centrifugation with 500× *g*, the cells were stored at −80 °C in 1 mL freezing medium containing 10 parts fetal calf serum (LINARIS, Dossenheim, Germany) and 1 part DMSO (Roth, Karlsruhe, Germany).

### 2.3. Preparation of Tissue Samples

All tissue samples were obtained from patients undergoing endonasal sinus surgery due to CRSwNP (*n* = 10) or CRSsNP (*n* = 8). Additionally, healthy nasal mucosa from the inferior turbinate was collected from patients without CRS undergoing septoplasty with turbinoplasty due to non-inflammatory reasons (*n* = 8). Tissue samples were dissociated using the Multi-Tissue Dissociation Kit and the gentleMACS Dissociator (Miltenyi Biotec GmbH, Bergisch Gladbach, Germany) according to the manufacturer’s protocols. Next, the cell suspension was filtered through a mesh (sizes 100 µm to 40 µm) together with 15 mL of RPMI solution (Biochrom GmbH, Berlin, Germany). After centrifugation with 500× *g*, cells were frozen at −80 °C in 1 mL freezing medium containing 10 parts fetal calf serum (LINARIS, Dossenheim, Germany) and 1 part DMSO (Roth, Karlsruhe, Germany).

### 2.4. Cell Sorting of CD4+, CD8+, and CD3− Antigen-Presenting Cells (APC)

Cell sorting was performed as positive selection of CD4+ and CD8+ T cells and negative selection of CD3− APC cells via MACS cell separation, with human-labeled CD3, CD4, and CD8 microbeads (Miltenyi Biotec GmbH, Bergisch Gladbach, Germany), according to the manufacturer’s protocol. APCs were only collected from peripheral blood. For this, only labelled CD3 microbeads were used, and cells that were not magnetically sorted were defined as CD3− APCs. APCs from the tissue were not used; therefore, no epithelial or stromal cells belonged to this population. Afterwards, CD4+ and CD8+ T cells were counted with a cell counter and analyzer system (CASY TT, Innovatis, Reutlingen). It was not possible to count CD3 negative antigen-presenting cells (APC) with the CASY cell counter in our lab. For this reason, the cell counting was carried out with a Neubauer counting chamber.

### 2.5. Cell Culture with C. albicans Peptide Mix for 6 Days

Cell cultures of CD4+ or CD8+ T cells from all tissue samples with *C. albicans*, a negative control, and a positive control were performed using a 96-well microplate. We added 2.5 × 10^4^ CD4+ or CD8+ T cells, 5 × 10^4^ APC, 200 µL TexMacs Medium supplemented with 1% Penicillin/Streptomycin (Miltenyi Biotec GmbH, Bergisch Gladbach, Germany), and 30 IU/mL IL-2 per well, together with 1 µg/mL of a commercially available peptide pool from *Candida albicans* (Peptivator *C. albicans* MP65 research grade, Miltenyi Biotec GmbH, Bergisch Gladbach, Germany), or 2 µL of human T-Activator CD3/CD28 Dynabeads (Miltenyi Biotec GmbH, Bergisch Gladbach, Germany), in order to obtain a bead-to-cell ratio of 1:1. Cultivation was performed for 6 days at 37 °C/5% CO_2_ on a round-bottom 96-well plate, and a medium change was not necessary. IL-2 was added to the cell culture in order to reach a higher cell viability in the experiments. Cell viability was measured using Viability Dye 780 (eBioscience; Thermo Fisher Scientific Inc., Waltham, MA, USA) during the FACS measurement. Due to a low amount of T cells in tissue samples from patients with CRSsNP and healthy donors, cell cultures were performed in some of these cases with lower cell amounts. After 6 days in culture, cells were separated from the supernatants by centrifugation (500× *g*, 5 min), and the dynabeads were removed by magnetic separation from the positive control.

### 2.6. Fluorescence-Activated Cell Sorting

T cell activation was determined by FACS, as previously described [20], by measuring the expression of Ki-67. Different antibodies were used for the gating strategy: anti-CD45 Pacific Orange (1:300, Thermo Fisher Scientific Inc., Waltham, MA, USA), anti-CD3 phycoerythrin (PE)-Cy7 (1:300), anti-CD4 Pacific Blue (1:50), anti-CD8a Alexa 700 (1:50) (all BioLegend, Inc., San Diego, CA, USA), and anti-Ki-67 (1:100, BD Bioscience, San Jose, CA, USA). Isotype control staining was performed using mouse-IgG PE (1:200, BD Bioscience, San Jose, CA, USA). Staining of apoptotic cells was performed using Viability Dye 780 (1:10, eBioscience; Thermo Fisher Scientific Inc., Waltham, MA, USA). After blocking with 25 μg/mL normal mouse immunoglobulin G (Sigma-Aldrich; Merck KGaA, Darmstadt, Germany) for 15 min on ice, cell surface staining was performed on ice for 30 min. For staining with the intracellular marker Ki-67, fixation buffer (eBioscience; Thermo Fisher Scientific Inc., Waltham, MA, USA) was added to the cells for 30 min at room temperature (RT), followed by anti-Ki-67 for 45 min at RT. All antibodies were used according to the manufacturers’ protocols. FACS analysis was performed using an LSR II flow cytometer (Becton, Dickinson and Company, San Diego, CA, USA). FlowJo software (FlowJo LLC, Ashland, OR, USA) was used for data analysis. First, all CD45+ lymphocytes were selected. Afterwards, all apoptotic cells were excluded by gating on CD3 and viability dye, followed by gating on CD4 or CD8 positivity. In the last step, gating was performed on Ki67 and CD4 or CD8.

### 2.7. Cytokine Secretion Measures in the Supernatant by Cytometric Bead Array

Cytokine secretion was performed using the cytometric bead array Legendplex™ Human Cytokine Panel (13-plex) (Biolegend, San Diego, CA, USA). The supernatants of the cell cultures were collected after 6 days and frozen at −80 °C until the multiplex ELISA was performed. According to the protocol, thawed supernatants were mixed with capture beads of different sizes and different levels of allophycocyanin fluorescence. Washing steps and addition of biotinylated detection antibodies and Streptavidin–phycoerythrin (SA–PE) followed. Afterwards, samples were measured using a FACS Canto (Becton, Dickinson and Company, San Diego, CA, USA). Analysis of the cytokine concentrations of IL-2, IL-4, IL-5, IL-6, IL-9, IL-10, IL-13, IL-17a, IL-17f, IL-21, IL-22, INF-γ, and TNF-α were performed using LegendplexTM data analysis software (Biolegend, San Diego, CA, USA).

### 2.8. Statistics

Data are presented as the means ± standard deviation (SD). Statistical significance was measured by one-way ANOVA using the Holm–Sidak test, paired t-test or, for non-parametric distribution, the Kruskal–Wallis test was applied using GraphPad Prism Software 6.0c. Values of *p* < 0.05 were considered to be statistically significant.

## 3. Results

### 3.1. Patient Characteristics

A total of 26 patients were included in the study. The mean age of all patients was 43.38 ± 18.61 years. Topical steroids (mometasone furoate nasal spray) were used by all patients prior to surgery. If the patients did not benefit from conservative therapy, indication was set for paranasal sinus surgery according to the European guidelines [1]. Surgery was performed between December 2017 and October 2018 at the same university’s ENT Department. Patients with Churg–Strauss syndrome, primary ciliary dyskinesia, or cystic fibrosis were excluded. Patient characteristics are summarized in Table 1.

### 3.2. Activation of CD4+ and CD8+ T Cells by C. albicans Antigen

To measure T cell responses towards *C. albicans* antigen we used the proliferation marker Ki-67. No differences in activated CD4+ T cells (9.67 ± 5.68% vs. 7.80 ± 4.97%) were observed between tissue and peripheral blood CD4+ T cells in patients with CRSsNP (Figure 1a). Ki-67 was significantly more expressed in CD8+ T cells from inflammatory nasal tissue compared to peripheral blood CD8+ T cells in these patients (22.82 ± 18.14 vs. 14.34 ± 15.00%, *p* = 0.039) (Figure 1b).

In patients with CRSwNP, no significant differences in the expression of Ki-67 were observed between tissue CD4+ (14.01 ± 8.32% vs. 9.75 ± 10.08%) or CD8+ T cells (23.63 ± 16.02% vs. 15.77 ± 11.57%) and peripheral blood CD4+ and CD8+ T cells, respectively (Figure 1c,d). The same was true for the control group: %Ki-67+/CD4+ T cells (14.52 ± 18.26% tissue vs. 13.99 ± 18.73% blood); %Ki-67+/CD8+ T cells (18.8 ± 23.12% tissue vs. 19.30 ± 17.21% blood) (Figure 1e,f).

Negative controls without antigen stimulus were also performed. The only significant difference was observed between CD8+ T cells from nasal tissue in comparison to peripheral blood in CRSsNP. In this case the basal proliferative potential of unstimulated CD8+ T cells, comparing cells from peripheral blood and nasal tissue, was 7.08 ± 5.25 (peripheral blood) vs. 13.59 ± 13.03 (nasal tissue).

Regarding the expression of Ki-67 among tissue-derived T cells stimulated with *C. albicans* antigen, no significant differences were found between patients with CRSwNP, CRSsNP, or the control group. This was true for both CD4+ (16.20 ± 7.77% vs. 9.67 ± 5.68% vs. 14.52 ± 18.26%) and CD8+ T cell responses (27.53 ± 15.56% vs. 22.82 ± 18.14% vs. 18.87 ± 23.12%) (Figure 2a,b). Likewise, CD4+ (11.12 ± 10.91% vs. 7.80 ± 4.97% vs. 13.99 ± 18.73%) and CD8+ T cells (19.16 ± 10.31% vs. 14.34 ± 14.10% vs. 19.30 ± 17.21) from the peripheral blood of patients and controls also did not respond differently to *C. albicans* antigen stimulation (Figure 2c,d).

### 3.3. No Differences in Cytokine Secretion of Tissue CD4+ and CD8+ T Cells

After 6 days in culture, the cytokine secretion of tissue CD4+ and CD8+ T cells was measured. Although CD8+ T cells in CRSsNP patients were significantly more activated in nasal tissue than in peripheral blood, no differences in cytokine secretion were found between these two groups (Figure 3).

Similarly, no differences in cytokine secretion after stimulation with *C. albicans* antigen were seen when comparing CD8+ T cells from the nasal mucosa of CRSwNP and CRSsNP patients. Comparing the cytokine secretion of tissue CD8+ T cells from all three study groups, significantly higher levels of IL-17a, IL-17f, and IL-9 were measured in the control group in comparison to tissue T cells from patients with CRSwNP (Figure 4). IL-17a was also significantly higher in the control group compared to CRSsNP in the CD8+ T cell culture (Figure 4). Regarding CD4+ T cells, no differences in cytokine secretion were measured.

### 3.4. Stratification of Patients into Low and High Responders Based on Tissue CD4+ and CD8+ T Cell Responses to C. albicans

Within the various study groups, a high variability in the expression of the activation marker Ki-67 between tissue samples was seen. Consequently, a subgroup analysis of non-responders and responders was performed for CD4+ (Figure 5A) and CD8+ (Figure 5B) T cells from patients with CRSwNP and CRSsNP. Differences in Ki-67 expression were observed (Figure 5). A significant difference (*p* = 0.004) between responders (20.50 ± 6.25) and non-responders (7.52 ± 3.41) was measured in CD4+ T cells from CRSwNP patients.

## 4. Discussion

To date, the role of fungi in the pathophysiology of CRS is unclear. A high colonization of the nasal cavity was found in several studies [18,21,22]. However, the question arises whether this colonization is part of the normal bacterial and fungal biofilm composition, or can be regarded as a special finding in patients with CRS. Interestingly, in a study published by Lackner et al., positive fungal cultures in 94% of healthy neonates in the first 4 months of life [21] were found. Newer studies have focused on molecular techniques to describe the fungal colonization of the nasal cavity [22]. No relevant differences were found between healthy controls and patients with chronic rhinosinusitis, but the observed fungi differed between the studies [23,24].

In an early study by Shin et al. [25], it was hypothesized that airborne fungi induce an abnormal immunological reaction in CRS patients. Thus, peripheral blood, but not local tissue T cells, was stimulated with *Alternaria*, *Aspergillus*, *Cladosporium*, and *Penicillium* in vitro. This hypothesis was later rejected by Fokkens et al., mainly due to differences to healthy controls [26].

The aim of this study was to evaluate whether the high colonization of the nasal cavity with fungi can influence the activation of local tissue T cells and lead to an ongoing inflammatory reaction in patients with CRS in comparison to healthy controls. In summary, a proliferation of *C. albicans*-specific tissue CD8+ T cells compared to peripheral blood CD8+ T cells was found exclusively in patients with CRSsNP. No differences were detected between local CD4+ and CD8+ tissue T cells when comparing CRSwNP and CRSsNP patients, as well as healthy controls. However, high interindividual differences were observed among patients with CRSwNP and CRSsNP in the expression of the activation marker Ki-67 after stimulation with *C. albicans* antigen. Patients with up to 50% Ki-67+ cells among tissue CD8+ T cells after stimulation with *C. albicans* antigen were found in both CRSwNP and CRSsNP. In both subgroups of CRS patients, a classification of low and high responders with respect to CD4+ and CD8+ T cell reactivity was performed. Significant differences were measured between the two groups of “responders” and “non-responders” of CD4+ T cells from CRSwNP patients. These results lead to the conclusion that at least this subgroup of patients with CRSwNP may develop an inflammatory reaction triggered by C. albicans antigen-activated T cells. However, our data cannot finally prove the existence of such subgroups. Thus, it will be necessary to perform experiments with higher numbers of participants in order to validate a subgroup of “responder” to *C. albicans* antigen. These results lead to the conclusion that a subgroup of patients with CRSwNP as well as CRSsNP may develop an inflammatory reaction triggered by *C. albicans* antigen-activated T cells. If these responders could be defined more precisely in future studies, a targeted special antifungal therapy may result in better outcomes for this subgroup of patients.

Activation of tissue CD8+ T cells was significantly higher than in peripheral blood CD8+ T cells in patients with CRSsNP, which suggests that CD8+ T cells specific to *C. albicans* might play a role in the pathogenesis of CRSsNP. In contrast to this finding, no differences were found between the different study groups CRSwNP, CRSsNP, and healthy controls. Negative controls without antigen stimulus were also performed. The only significant difference was observed between CD8+ T cells from nasal tissue and those from peripheral blood in CRSsNP. In this case, the basal proliferative potential of unstimulated cells comparing to cells from peripheral blood and nasal tissue was 13.59 ± 13.03 vs. 7.08 ± 5.25. However, these results do not affect our conclusion that differences were observed in patients with CRSsNP. To our estimation, non-stimulated cells would be only useful in comparisons within the groups of tissue-derived T cells or peripheral blood T cells.

A unique role of Th17 cells and their cytokines—especially IL-17A, IL-17F, and IL-22—has been described [10]. In the present study, no differences were found in the secretion of cytokines between the study groups. Quite contrary to the expected results, a significantly higher secretion of IL-17A and IL17F was found in CD8+ T cells of the healthy controls.

Nevertheless, this study has certain limitations. First, each patient with CRSwNP and CRSsNP received a conservative medical therapy with topical steroids prior to surgery. This may have influenced the results due to its immunosuppressive character, potentially leading to a reduced response of the T cells to the antigen stimulus, especially with respect to cytokine secretion by activated T cells. This is a possible explanation for the fact that differences were observed in proliferation, but not in cytokine secretion, between tissue and peripheral blood CD8+ T cells. Consequently, a higher cytokine secretion in the control group is also possible, since this group did not receive any steroid therapy preoperatively. Second, as a pilot study, the number of samples was low. High interindividual differences were observed, and it seems that patients can be divided into low and high responders to the stimulus of *C. albicans* antigens. This should be addressed in future studies. In the tissue from some groups, a very low amount of T cells could be identified, especially in the nasal mucosa from healthy controls. This fact must be taken into account when critically discussing cytokine levels in our tests. The low cell density due to restricted cell amounts affects the measurement of cytokines in the supernatant.

Many published studies have linked indoor exposure to mold and water dampness to sinusitis. A meta-analysis of 31 published studies reported an increased risk for rhinitis and its subtypes in relation to mold odor, visible odor, or exposure to dampness [27]. In addition, these findings were approved by a meta-analysis of 12 published studies that reported an improvement in wheezing and rhinitis through professional remediation of mold- and water-damaged homes [28]. These findings provide epidemiological evidence linking fungal exposure to chronic rhinosinusitis. Most studies in the literature dealing with the role of fungi focused on peripheral blood T cells. To the best of our knowledge, no study evaluated the reactivity towards the *C. albicans* (peptide) antigen in a long-term cell culture with tissue-derived T cells from patients with CRS. These results are in accordance with other studies that rejected the hypothesis that fungi influence CRS [26,29]. Nevertheless, high interindividual differences in every study group were found. A possible explanation as to why antifungal medication does not affect the outcome in CRS patients is that the *C. albicans*-specific T cells might be tissue-resident [3], and therefore would not need any antigen stimulus to produce enhanced levels of cytokines in the nasal mucosa. However, the antigen specificity of up to 50% of nasal T cells might be a reasonable basis for antigenic therapy similar to specific immunotherapy used, for example, in wasp venom allergy [30].

## Figures and Tables

**Figure 1 jof-07-00403-f001:**
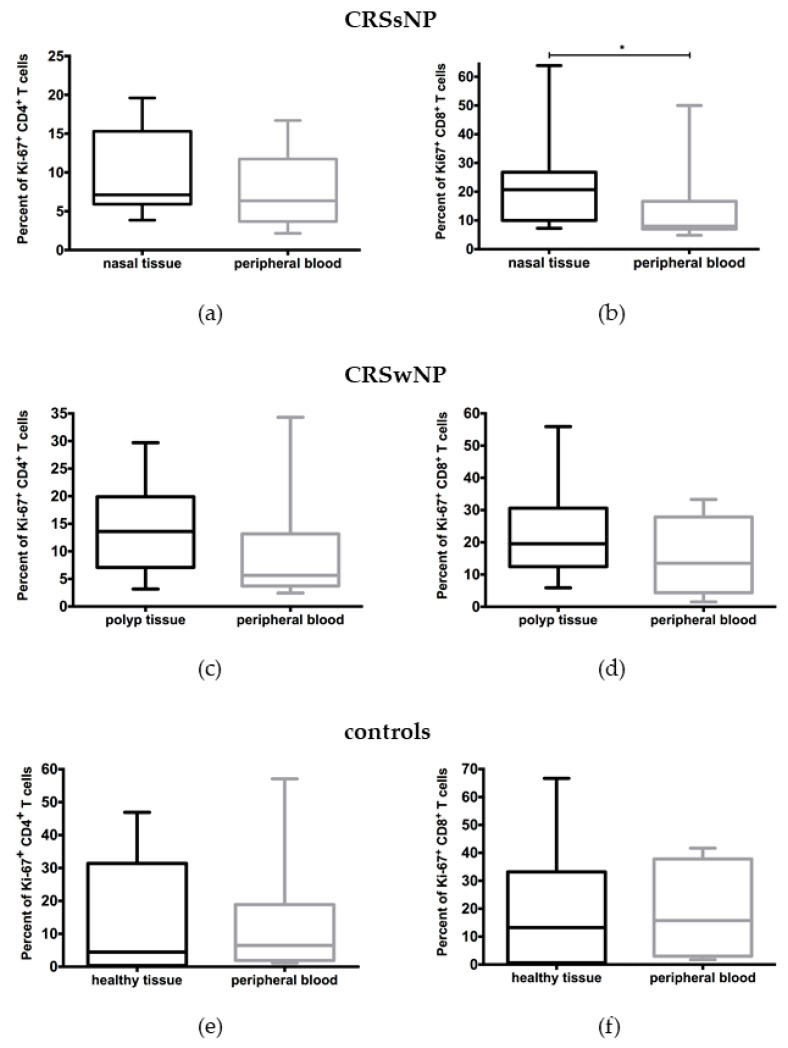
Stimulation of T cells by a peptide pool of *Candida albicans* after 6 days in culture. (**a**,**b**) demonstrate data from CRSsNP patients; (**c**,**d**) demonstrate data from CRSwNP; and (**e**,**f**) demonstrate data from healthy controls. No significant differences were found in the expression of Ki-67 in comparison between tissue and peripheral blood CD4+ T cells from patients with (**a**) CRSsNP, (**c**) CRSwNP, or (**e**) healthy controls. (**b**)A significantly higher activation of tissue CD8+ T cells in comparison to peripheral blood CD8 T cells was found in patients with CRSsNP. No differences were found regarding CD8+ T cells in (**d**) patients with CRSwNP or in (**f**) healthy controls. * *p* < 0.05.

**Figure 2 jof-07-00403-f002:**
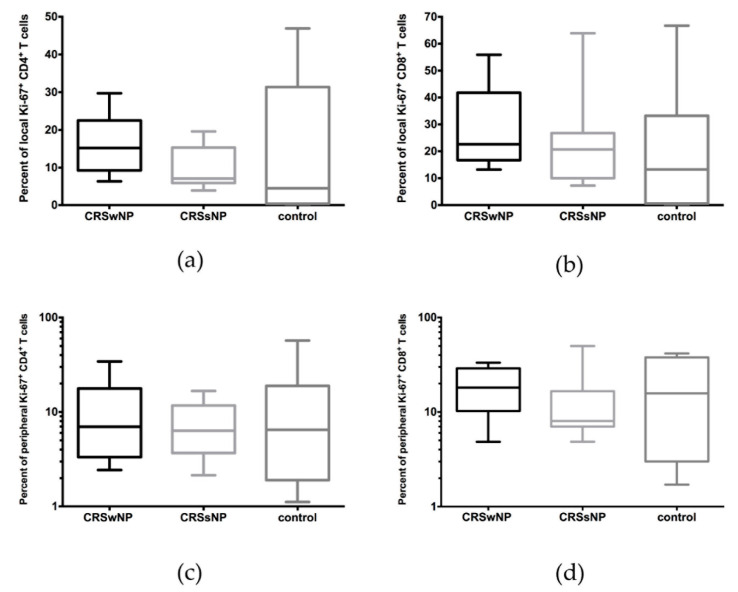
Comparison of (**a**,**b**) local and (**c**,**d**) peripheral CD4+ and CD8+ T cells among the patients with CRwNP, CRSsNP, and healthy controls, after stimulation with a peptide pool of *C. albicans* over 6 days. No significant differences were observed.

**Figure 3 jof-07-00403-f003:**
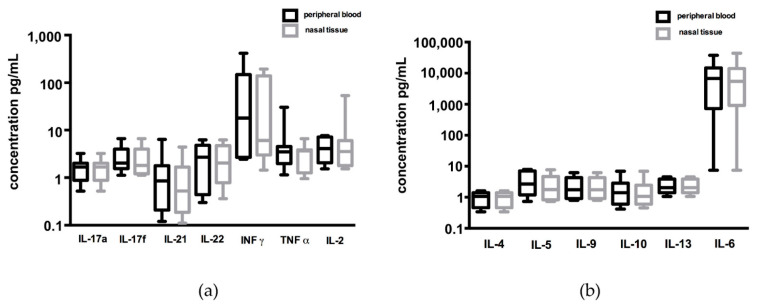
Comparison of cytokine secretion by CD8+ T cells from peripheral blood and inflammatory nasal tissue in CRSsNP patients: high levels of INFγ and IL-6 were found in both groups, but without any significance between both groups regarding (**a**) Th1/Th17 and (**b**) Th2 cytokines. T cells were stimulated with a peptide pool of C. albicans for 6 days, and cytokines were measured in the supernatant of the cell culture after 6 days.

**Figure 4 jof-07-00403-f004:**
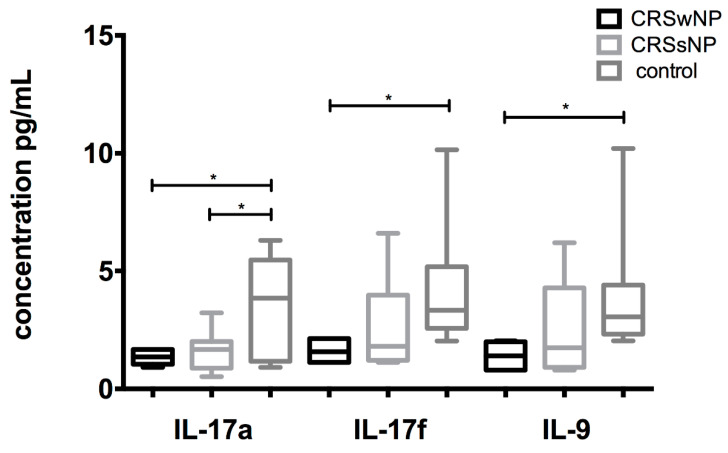
Comparison of IL-17a, IL-17f, and IL-9 secretion between patients with CRSwNP, CRSsNP, and healthy individuals: A significantly higher secretion of IL-17f and IL-9 was found in the supernatant in the local CD8+ T cells from the healthy controls in comparison to the local CD8+ T cells from CRSwNP patients. A higher concentration of IL-17a was found in the healthy controls in comparison to both subtypes of CRS. * *p* < 0.05.

**Figure 5 jof-07-00403-f005:**
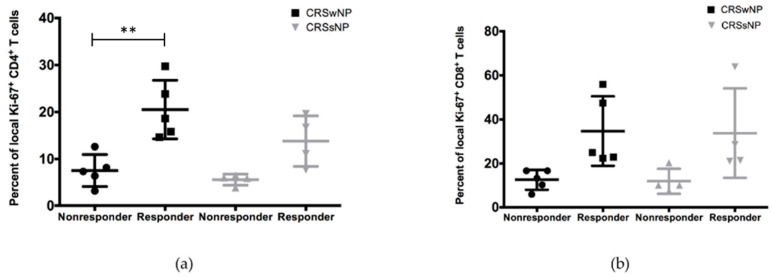
Subgroup analysis of local (**a**) CD4+ and (**b**) CD8+ T cells from patients with CRSwNP and CRSsNP based on their expression of Ki-67. Significant differences were measured between CD4+ T cells from CRSwNP.

**Table 1 jof-07-00403-t001:** Patients’ characteristics.

Study Group	Age, Years (±SD)	Sex, Female/Male	Allergy, *n* (%)	Previous Surgery, *n* (%)
CRSwNP (*n* = 10)	51 (±19)	2/8	4 (40)	5 (50)
CRSsNP (*n* = 8)	40 (±15)	3/5	4 (50)	1 (12.5)
Control group (*n* = 8)	35 (±18)	0/8	5 (62.5)	2 (25)

## Data Availability

The data presented in this study are available on request from the corresponding author. The data are not publicly available due to ethical issues.

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
