# Peer review of "Detection of Candida albicans-Specific CD4+ and CD8+ T Cells in the Blood and Nasal Mucosa of Patients with Chronic Rhinosinusitis"

_jof, 2021, doi:10.3390/jof7060403_

Round 1

Reviewer 1 Report

I have few major comments and the rest are minor remarks. I think this study has been well performed, but some more information of the background and some methodological details need to be improved. 

Major points:

-Method section: a paragraph describing how the "peptide pool derived from C. albicans" has been isolated and quantified is completely missing. The author should add it in order to ensure the replicability of the study

-Cell viability after 6 days: in our experience lymphocytes needs serum to survive for 6 days in cell culture, otherwise, the cell viability will be severely impaired. IL-2 would indeed help, but was the medium partially changed at a certain point during the 6 days? The author should prove that the cell viability was not affected using this specific cell medium (that does not seem to contain serum). Ki67 is an index of proliferation but does not give information on the basal viability of the cell.

-Line 91: why some cell type have been counted with Neubauer counting chamber and other with CASY?

-Line 104: "Due to a low amount of T cells in tissue samples from patients with CRSsNP and healthy donors, cell cultures were performed in some of these cases with lower cell amounts." . Doesn't this imply (unless the data have been normalized for the number of cells in the well) that the statistical comparison between the peripheral cells and the nasal-derive cells might be affected by the cell differential counts? The author should address this issue 

-Do the authors have a negative control with cells that have not been stimulated with Candida peptides? How is the basal proliferative potential in unstimulated cells when comparing cells from peripheral blood and nasal tissue?

-Line 213: 'cytokine expression of the T cells was not affected (unpublished data).' : if this result is mentioned here it should be provided in the manuscript, otherwise this sentence has to be removed. Why the authors do not provide those data?

Minor points:

-please write all the microorganism names in italic: e.g. Candida albicans

-in the abstract, the acronym CRS appears without having specified that it means Chronic rhinosinusitis, and the same is for CRSsNP. People reading only the abstract might not be following.

-Line 109-112: please add which dilution of antibodies have been used to ensure the replicability of the study, as well as the gating strategy of the FACS analysis.

- why the authors use the term "edaphic"? I was wondering if this term commonly used in medicine when talking about mucosal cells? or more for plant biology?

Author Response

Manuscript Ickrath et al.: “Detection of Candida albicans-specific CD4+ and CD8+ T cells in blood and nasal mucosa of patients with chronic rhinosinusitis”

Manuscript ID: jof-1190818

Dear editor,

Thank you very much for giving us the possibility to revise the above mentioned MS in order to be publishable in Journal of Fungi. The referees’ comments proved to be very helpful to improve the MS and so we would like to submit a list of changes in response to the reviewers’ remarks. As we do receive an internal funding by the Open Access Publication Fund of the University of Wuerzburg if the manuscript will be accepted, I additionally added this information to Acknowledgments of the manuscript: "This publication was supported by the Open Access Publication Fund of the University of Wuerzburg."

Reviewer #1:

  • Method section: a paragraph describing how the "peptide pool derived from C. albicans" has been isolated and quantified is completely missing. The author should add it in order to ensure the replicability of the study

We used a peptid pool from Candida albicans, which is distributed by Miltenyi Biotec (Peptivator C. albicans MP65 research grade, Miltenyi Biotec GmbH, Bergisch Gladbach, Germany). According to the manufacturers’ protocol, the 65 kDa mannoprotein MP65 from the cell wall is described as a major T cell antigen and can be used for antigen-specific T cell stimulation. Since this product is commercially available, it can be used reproducibly for further studies. We specified this information in the methods section. (Line 109)

  • Cell viability after 6 days: in our experience lymphocytes needs serum to survive for 6 days in cell culture, otherwise, the cell viability will be severely impaired. IL-2 would indeed help, but was the medium partially changed at a certain point during the 6 days? The author should prove that the cell viability was not affected using this specific cell medium (that does not seem to contain serum). Ki67 is an index of proliferation but does not give information on the basal viability of the cell.

We have had the same experiences with our cell cultures. In a preliminary examination, we had initially focused on determining cell viability by a viability dye in our FACS panel. Due to a high rate of apoptotic cells, we added IL-2 to our cell cultures reaching a sufficient number of living cells. We used serum-free TexMacs Medium supplemented with 1% Penicillin/Streptomycin (Miltenyi Biotec GmbH, Bergisch Gladbach, Germany) and a medium change was not necessary. Cell viability was measured by Viability Dye 780 (eBioscience; Thermo Fisher Scientific Inc., Waltham, MA, USA) in our FACS panel. We added this information to the methods section. (Line 114-116)

  • Line 91: why some cell type have been counted with Neubauer counting chamber and other with CASY?

It is not possible to count CD3 negative antigen presenting cells (APC) with the CASY cell counter in our lab. For this reason, the cell counting was carried out with the Neubauer counting chamber. The protocol for CD4+ and CD8+ T cells with the CASY in our lab is reliable. We added this information to the methods section. (Line 100-103)

  • Line 104: "Due to a low amount of T cells in tissue samples from patients with CRSsNP and healthy donors, cell cultures were performed in some of these cases with lower cell amounts." Doesn't this imply (unless the data have been normalized for the number of cells in the well) that the statistical comparison between the peripheral cells and the nasal-derive cells might be affectted by the cell differential counts? The author should address this issue

Thank you for this advice. Usually we find lower cell amounts in tissue samples from healthy controls and from CRsNP patients in comparison to tissue samples from CRSwNP patients. This problem occurs only in tissue samples and not in peripheral cells. We added this limitation to the discussion section. (Line 309-313)

  • Do the authors have a negative control with cells that have not been stimulated with Candida peptides? How is the basal proliferative potential in unstimulated cells when comparing cells from peripheral blood and nasal tissue?

We also performed negative controls. The only significant difference was observed between CD8+ T cells from nasal tissue in comparison to peripheral blood in CRSsNP. In this case, the basal proliferative potential of unstimulated cells comparing cells from peripheral blood and nasal tissue was 7.08 ± 5.25 (peripheral blood) vs. 13.59 ± 13.03 (nasal tissue). (Line 192-196)

  • Line 213: 'cytokine expression of the T cells was not affected (unpublished data).' : if this result is mentioned here it should be provided in the manuscript, otherwise this sentence has to be removed. Why the authors do not provide those data?

Thank you for this advice. Since this sentence does not provide any additional information, it was removed from the manuscript. Cytokine secretion was nearly the same between responder and non-responder. Due to the low number of experiments in every group, statistical analysis would not have a strong impact and was not performed.

  • please write all the microorganism names in italic: e.g. Candida albicans

Thank you for this advice. The microorganism names are now written in italics.

  • in the abstract, the acronym CRS appears without having specified that it means Chronic rhinosinusitis, and the same is for CRSsNP. People reading only the abstract might not be following.

Chronic rhinosinusitis (CRS) and chronic rhinosinusitis without nasal polyps (CRsNP) was added to the abstract. (Line 13, 16)

  • Line 109-112: please add which dilution of antibodies have been used to ensure the replicability of the study, as well as the gating strategy of the FACS analysis.

The dilution of the antibodies were added to the manuscript (Line 125-129). First, all CD45+ lymphocytes were selected. Afterwards, all apoptotic cells were excluded by gating on CD3 and viability dye followed by gating on CD4 or CD8 positivity. In the last step, gating was performed on Ki67 and CD4 or CD8. (Line 138-140)

  • why the authors use the term "edaphic"? I was wondering if this term commonly used in medicine when talking about mucosal cells? or more for plant biology?

„Edaphic“ was changed to tissue T cells in the manuscript, which is more precise.

Dear editor, if further data are needed to improve the MS, we will be happy to follow your suggestions. 

Yours sincerely,

Pascal Ickrath

Reviewer 2 Report

The aim of this study was to address a potential role for C. albicans-specific T cells in CRS by analyzing T cells from human tissue explants from patients with CRS with and without polyps, and comparing them to T cells from healthy individuals regarding T cell proliferation and production of cytokines. The use of human samples is a strength of this manuscript. However, the lack of a comparison between stimulated (peptide) vs non-stimulated (control) cells is a problem because the reader cannot tell whether activation of T cells by the peptide is taking place. Also, it is not clear if antigen-presenting cells were used in the experiments. Moreover, TCRs from T cells are not supposed to recognize peptides that are not conjugated to a MHC molecule, meaning that if there were no APCs in the system, or immobilized MHC-peptide complexes, those T cells were not supposed to activate by the presence of the C. albicans-derived peptide.

Abstract

CRS: first time cannot be abbreviated

CRSsNP: cannot be abbreviate first time

I missed the link between C. albicans and CRS – the rationale behind the manuscript

Intro

Line 40: initiated is more adequate than initialized

Line 41: candidiasis, no capital letter

Line 51: interleukin, no capital letter

Line 51: which fungus? Candida or any fungus? Please clarify

Line 58: “The aim…was TO MEASURE….”

Methods – Confusing in general. Other groups would struggle to reproduce the experiments due to vague description

Line 66: Intraopertiv

Line 69: equal amounts of Ficoll… and blood?

Line 82: sizes 40 to 100 um, or 100 nm to 40 um?

Line 87/88: Does negative selection of CD3- cells mean that the cells that were not magnetically sorted by CD4 and CD8 beads were recovered? If yes, aren’t they CD3-, CD4- and CD8-? If other method was used, please clarify. Are all the CD3- cells from the tissue, APCs? Are there epithelial or stromal cells in this tissue? This section is confusing, consider rewriting.

Line 88: ”…human CD3+, CD4+ and CD8+ Microbeads” – Cells can be positive or negative for antigens, beads cannot.

Line 95: “2.5 x 104 CD4+ or CD8+ T cells, 5 x 104 APC” – Scientific notation has to be fixed

Line 128: APC was used before for “antigen-presenting cell”, here is probably allophycocyanin. Readers can be misled by this. Please clarify

Results

Line 153 – Consider to present the proliferation kinetic as supplementary material

Figure 1: Panel has letter ”e” twice. “nasal tissue”, “polyp tissue”,  and “healthy tissue” should be changed to CRSsNP, CRSwNP and control, respectively.

Fig 1 legend: “Stimulation of T cells by Candida albicans after 6 days in culture”. It suggests the use of living yeast, rather than the mp65-derived antigen? Legend headline is confusing.

Most of the times when the authors show numbers in the text, is it difficult to know exactly what groups are being compared

Line 171: “The negative control (cultivation without C. albicans antigen) of each sample showed a significantly lower expression of Ki-67 in comparison to the positive control” – I could not find these results. It would help the reader if they were in Fig 1, to show the activation of T cells by C.albicans’ antigen

Figure 2: If control means cells from healthy individuals, a negative control (cells without antigen) from each group should be added. If a positive control (cells + anti-cd3/cd28) was done, it could be included too. Also, I could not find for how long the cells were stimulated with C. albicans’ antigen in the experiment described in Fig. 2

Line 185: “No differences in cytokine secretion of edaphic CD4+ and CD8+ T cells”. No data regarding CD4+ cells was showed in this section and in Figure 3

Figure 3: The legend is vague. A figure and its legend should be enough for the reader to understand the result. Also, in my opinion non-stimulated cells should be showed.

Figure 4: Same as Fig. 3, assuming that control group is composed by healthy individuals rather than CRS cells without antigen

Line 213: Avoid citing ”data not shown”, it is better to show it as supplementary data than not showing it at all

Figure 5: Can at least a t-test comparing the CRSwNP groups be made?

Discussion

Lines 225-227: healthy ones or neonates with CRS?

Lines 238-240: Enrichment of CD8+ cells was not addressed in the results. Activation or proliferation of CD8+ cells were showed in Fig. 1

Line 246: It suggests that living C. albicans was used to stimulate the cells, instead of the antigen.

Lines 247-250: I disagree. Unfortunately, without some statistical analysis no conclusion can be taken from the subgrouping strategy

Line 251: C. albicans or antigen from C. albicans?

Lines 250-251: I disagree. To state that “a subgroup of patients with CRSwNP as well as CRSsNP may develop an inflammatory reaction triggered by C. albicans-activated T cells” the authors should have shown non-stimulated vs stimulated cells from each group and performed a statistical analysis showing differences. Also, Figure 4 shows that CD8+ T cells from healthy individuals produced more IL17, IL-17F and IL-9 than cells from CRS patients. All these cytokines have inflammatory properties although in distinct contexts. Based on the results presented on this manuscript the conclusion regarding an inflammatory feature of C. albicans’ antigen-activated T cells cannot be taken.

Lines 255-257: Maybe this sentence could be moved to Introduction.

Lines 257-260: I disagree. Without non-stimulated cells is impossible to say that there was any activation at all.

Line 271: The fact that fewer T cells from some groups were used (as mentioned in lines 102-104) can also explain why the cytokines were not as expected. If the cell density (cells/mL) was kept even with a lower number of cells, reliable results could be expected. However if the cell density was lowered due to fewer cells, the measurement of cytokines in the supernatant could be impacted. This hypothesis does not exclude the suppression due to corticoid treatment. Also, if non-stimulated cells (negative controls) are not showed in Results, it is hard for the reader to judge whether the T cells were activated at all, or if the cells were in any way responsive, or if the C. albicans antigen was active/non-degraded.

Line 277: Antigen or living yeast?

Line 279: non-infectious? Would it be uninfected?

Author Response

Manuscript Ickrath et al.: “Detection of Candida albicans-specific CD4+ and CD8+ T cells in blood and nasal mucosa of patients with chronic rhinosinusitis”

Manuscript ID: jof-1190818

Dear editor,

Thank you very much for giving us the possibility to revise the above mentioned MS in order to be publishable in Journal of Fungi. The referees’ comments proved to be very helpful to improve the MS and so we would like to submit a list of changes in response to the reviewers’ remarks. As we do receive an internal funding by the Open Access Publication Fund of the University of Wuerzburg if the manuscript will be accepted, I additionally added this information to Acknowledgments of the manuscript: "This publication was supported by the Open Access Publication Fund of the University of Wuerzburg."

Reviewer #2:

The aim of this study was to address a potential role for C. albicans-specific T cells in CRS by analyzing T cells from human tissue explants from patients with CRS with and without polyps, and comparing them to T cells from healthy individuals regarding T cell proliferation and production of cytokines. The use of human samples is a strength of this manuscript. However, the lack of a comparison between stimulated (peptide) vs non-stimulated (control) cells is a problem because the reader cannot tell whether activation of T cells by the peptide is taking place. Also, it is not clear if antigen-presenting cells were used in the experiments. Moreover, TCRs from T cells are not supposed to recognize peptides that are not conjugated to a MHC molecule, meaning that if there were no APCs in the system, or immobilized MHC-peptide complexes, those T cells were not supposed to activate by the presence of the C. albicans-derived peptide.

We also performed negative controls. The only significant difference was observed between CD8+ T cells from nasal tissue in comparison to peripheral blood in CRSsNP. In this case, the basal proliferative potential of unstimulated cells comparing cells from peripheral blood and nasal tissue was 13.59 ± 13.03 vs. 7.08 ± 5.25.

In every cell culture, APCs derived from peripheral blood of the same patient were added to the system additionally. Because of that, a T cell activation in the presence of C.albicans-derived peptides was possible.

  • CRS: first time cannot be abbreviated

CRS was changed to chronic rhinosinusitis (CRS) (line 13)

  • CRSsNP: cannot be abbreviate first time

CRSsNP was changed to chronic rhinosinusitis without nasal polyps. (line 16)

  • I missed the link between C. albicans and CRS – the rationale behind the manuscript

Candida albicans is ubiquitously present and colonization in the nose and oral cavity is common. In healthy patients they usually do not act as pathogens, but in some cases they can cause diseases. The influence of C. albicans as a trigger of T cell activation on the pathogenesis of chronic rhinosinusitis is discussed controversially and its exact role is not clear up to date. The aim of the present study was to measure the influence of C. albicans on the activation of CD4+ and CD8+ T cells in patients with CRS with and without nasal polyps. We added this information to the abstract. (line 10-12)

  • Line 40: initiated is more adequate than initialized

The words are changed in the main text (line 42)

  • Line 41: candidiasis, no capital letter

Correction was done (line 44)

  • Line 51: interleukin, no capital letter

Correction was done (line 57)

  • Line 51: which fungus? Candida or any fungus? Please clarify

In the cited study (Reference No. 11) extracts of Candida parapsilosis and Rhodotorula mucilaginosa were used for stimulation. We added this information to the main text. (line 57-58)

  • Line 58: “The aim…was TO MEASURE….”

Correction was done (line 64)

  • Line 66: Intraopertiv

Correction to “intraoperative” (line 72-73)

  • Line 69: equal amounts of Ficoll… and blood?

Senctence was changed to: “equal amounts of Ficoll (Biochrom, Berlin, Germany) and equal amounts of peripheral blood” (line 75-76)

  • Line 82: sizes 40 to 100 um, or 100 nm to 40 um?

Cell meshes with sizes from 100 µm to 40 µm were used, starting with larger pore sizes (100 µm). (line 88)

  • Line 87/88: Does negative selection of CD3- cells mean that the cells that were not magnetically sorted by CD4 and CD8 beads were recovered? If yes, aren’t they CD3-, CD4- and CD8-? If other method was used, please clarify. Are all the CD3- cells from the tissue, APCs? Are there epithelial or stromal cells in this tissue? This section is confusing, consider rewriting.

APCs were only collected from peripheral blood. We used only labelled CD3 Microbeads and defined the cells that were not magnetically sorted as CD3- APCs. We did not use APC from the tissue, therefore no epithelial or stromal cells belong to this population. We added this information to the main text. (Line 96-99)

  • Line 88: ”…human CD3+, CD4+ and CD8+ Microbeads” – Cells can be positive or negative for antigens, beads cannot.

We changed this sentence to “labelled CD3, CD4 and CD8 microbeads”. (Line 94-95)

  • Line 95: “2.5 x 104 CD4+ or CD8+ T cells, 5 x 104 APC” – Scientific notation has to be fixed

It was changed to: “2.5 x 104 CD4+ or CD8+ T cells, 5 x 104 APC” (Line 106-107)

  • Line 128: APC was used before for “antigen-presenting cell”, here is probably allophycocyanin. Readers can be misled by this. Please clarify

Thanks for this advice: APC was changed to allophycocyanin. (Line 146)

  • Line 153 – Consider to present the proliferation kinetic as supplementary material

Since this sentence does not provide any additional information, it was removed from the manuscript.

  • Figure 1: Panel has letter ”e” twice. “nasal tissue”, “polyp tissue”, and “healthy tissue” should be changed to CRSsNP, CRSwNP and control, respectively.

Thank you for this advice, the second (e) was changed into (f). In this figure (a-b) demonstrates data from CRSsNP patients, (c-d) demonstrates data from CRSwNP and (e-f) demonstrates data from controls. We modified figure 1 in order to clarify the provided information. (line 177)

  • Fig 1 legend: “Stimulation of T cells by Candida albicans after 6 days in culture”. It suggests the use of living yeast, rather than the mp65-derived antigen? Legend headline is confusing.

Peptide pool of Candida albicans was added to the figure legend. (line 178-179)

  • Line 171: “The negative control (cultivation without C. albicans antigen) of each sample showed a significantly lower expression of Ki-67 in comparison to the positive control” – I could not find these results. It would help the reader if they were in Fig 1, to show the activation of T cells by C.albicans’ antigen.

This information was not shown within the figures in order not to make them too confusing. Since these values are not shown, we have removed this sentence.

  • Figure 2: If control means cells from healthy individuals, a negative control (cells without antigen) from each group should be added. If a positive control (cells + anti-cd3/cd28) was done, it could be included too. Also, I could not find for how long the cells were stimulated with C. albicans’ antigen in the experiment described in Fig. 2

We also performed negative controls without antigen. The only significant difference was observed between CD8+ T cells from nasal tissue in comparison to peripheral blood in CRSsNP. In this case, the basal proliferative potential of unstimulated cells comparing cells from peripheral blood and nasal tissue was 7.08 ± 5.25 (peripheral blood) vs. 13.59 ± 13.03 (nasal tissue). In all experiments, stimulation with C. albicans antigen was done for 6 days. (Line 192-196, 207)

  • Line 185: “No differences in cytokine secretion of edaphic CD4+ and CD8+ T cells”. No data regarding CD4+ cells was showed in this section and in Figure 3

No significant differences were found regarding CD4+ T cells. For this reason, we did not show a special figure in this section. However, we added this missing information in the main text. (Line 225)

  • Figure 3: The legend is vague. A figure and its legend should be enough for the reader to understand the result. Also, in my opinion non-stimulated cells should be showed.

Since we saw no significant differences between peripheral blood and tissue T cells, we skipped measurements on non-stimulated cells. The figure legend in the main text has been adapted and formulated more precisely: “Comparison of cytokine secretion by CD8+ T cells from peripheral blood and inflammatory nasal tissue in CRSsNP patients: high levels of INFγ and IL-6 were found in both groups but without any significance between both groups regarding (a) Th1/Th17 and (b) Th2 cytokines. T cells were stimulated with a peptide pool of C. albicans for 6 days and cytokines were measured in the supernatant of the cell culture after 6 days.” (line 2-15-219)

  • Figure 4: Same as Fig. 3, assuming that control group is composed by healthy individuals rather than CRS cells without antigen

Our first aim was to demonstrate the influence of C. albicans in T cells from healthy individuals compared to CRS patients. That is why we chose that control group. The figure legend in the main text has been adapted and formulated more precisely: “ Comparison of IL-17a, IL-17f and IL-9 secretion between patients with CRSwNP, CRSsNP and healthy individuals: A significantly higher secretion of IL-17f and IL-9 was found in the supernatant in the local CD8+ T cells from the healthy in comparison to local CD8+ T cells from CRSwNP patients. A higher concentration of IL-17a was found in healthy controls in comparison to both subtypes of CRS. (line 228-232)

  • Line 213: Avoid citing ”data not shown”, it is better to show it as supplementary data than not showing it at all

Since this sentence does not provide any additional information, it was removed from the manuscript.

  • Figure 5: Can at least a t-test comparing the CRSwNP groups be made?

As recommended a t-test was made showing a significant difference (p=0.004) between responder (20.50 ± 6.25) and non-responder (7.52 ± 3.41) of CD4+ T cells from CRSwNP patients. We added this information to the main text and figure 5. (Line 239-240, 243, 270-277)

  • Lines 225-227: healthy ones or neonates with CRS?

Lackner et al. describe the fungal colonization of the nasal cavity in healthy neonates. (Line 250)

  • Lines 238-240: Enrichment of CD8+ cells was not addressed in the results. Activation or proliferation of CD8+ cells were showed in Fig. 1

“Enrichment” was changed to “proliferation”. (line 262)

  • Line 246: It suggests that living C. albicans was used to stimulate the cells, instead of the antigen.

Thank you for this advice, antigen was added to this sentence. (line 270)

  • Lines 247-250: I disagree. Unfortunately, without some statistical analysis no conclusion can be taken from the subgrouping strategy

As recommended, a t-test was made showing a significant difference (p=0.004) between responder (20.50 ± 6.25) and non-responder (7.52 ± 3.41) of CD4+ T cells from CRSwNP patients. We added this information to the main text and adapted our conclusion: “In both subgroups of CRS patients, a classification of low and high responders with respect to CD4+ and CD8+ T cell reactivity was performed. Statistically significant differences were only observed between responders and non-responders of CD4+ T cells from CRSwNP patients.” (Line 270-277)

  • Line 251: C. albicans or antigen from C. albicans?

“Antigens” was added to this sentence. (line 269)

  • Lines 250-251: I disagree. To state that “a subgroup of patients with CRSwNP as well as CRSsNP may develop an inflammatory reaction triggered by C. albicans-activated T cells” the authors should have shown non-stimulated vs stimulated cells from each group and performed a statistical analysis showing differences. Also, Figure 4 shows that CD8+ T cells from healthy individuals produced more IL17, IL-17F and IL-9 than cells from CRS patients. All these cytokines have inflammatory properties although in distinct contexts. Based on the results presented on this manuscript the conclusion regarding an inflammatory feature of C. albicans’ antigen-activated T cells cannot be taken.

In fact, we also do not think that C. albicans has a relevant inflammatory impact on T- cells in our complete cohort. However, there may be a subgroup of “responders” which can only be identified by a larger number of participants. We changed the sentence in order to weaken the meaning of our statement: “Significant differences were measured between the two groups of “responders” and “non-responders” of CD4+ T cells from CRSwNP patients. These results lead to the conclusion that at least this subgroup of patients with CRSwNP may develop an inflammatory reaction triggered by C. albicans antigen-activated T cells. However, our data cannot finally prove the existence of such subgroups. Thus, it will be necessary to perform experiments with higher numbers of participants in order to valid a subgroup of “responder” to C. albicans antigen” (line 270-277)

  • Lines 255-257: Maybe this sentence could be moved to Introduction.

This sentence was moved to the introduction.

  • Lines 257-260: I disagree. Without non-stimulated cells is impossible to say that there was any activation at all.

We also performed negative controls without antigen. The only significant difference was observed between CD8+ T cells from nasal tissue in comparison to peripheral blood in CRSsNP. In this case, the basal proliferative potential of unstimulated cells comparing to cells from peripheral blood and nasal tissue was 13.59 ± 13.03 vs. 7.08 ± 5.25. However, these results do not affect our conclusion, that differences were observed in patients with CRSsNP. To our opinion, non-stimulated cells would be only useful in the comparison within the group of tissue-derived T cells or peripheral blood T cells. (line 286-293)

  • Line 271: The fact that fewer T cells from some groups were used (as mentioned in lines 102-104) can also explain why the cytokines were not as expected. If the cell density (cells/mL) was kept even with a lower number of cells, reliable results could be expected. However if the cell density was lowered due to fewer cells, the measurement of cytokines in the supernatant could be impacted. This hypothesis does not exclude the suppression due to corticoid treatment. Also, if non-stimulated cells (negative controls) are not showed in Results, it is hard for the reader to judge whether the T cells were activated at all, or if the cells were in any way responsive, or if the C. albicans antigen was active/non-degraded.

Thank you for this information. That is a limitation of our study, which will be addressed in the discussion. We added following sentence as a possible limitation to our study: “In the tissue from some groups, a very low amount of T cells could be identified, especially in the nasal mucosa from healthy controls. This fact must be taken into account when critically discussing cytokine levels in our tests. The low cell density due to restricted cell amounts affects the measurement of cytokines in the supernatant. (line 309-313)

  • Line 277: Antigen or living yeast?

Antigen was added to the main text. (line 309)

  • Line 279: non-infectious? Would it be uninfected?

“Non-infectious” was deleted due to the above recommended corrections.

Dear editor, if further data are needed to improve the MS, we will be happy to follow your suggestions. 

Yours sincerely,

Pascal Ickrath

Reviewer 3 Report

                                                      4-29-2021

Review of Detection of Candida-albicans specific CD4+ and CD8+ Cells in Blood and Nasal Mucosa

Sinusitis is an all too common problem which often has significant morbidity.  The relationship between mold exposure and mold carriage/infection and sinusitis is somewhat controversial in some circles.

This paper has done some interesting research on CD4+ and CD8+ in blood and nasal mucosa on a fairly small set of chronic rhinosinusitus patients.  The paper appears to be well written, interesting, and well referenced.  I think it will be a useful addition to the literature.  I have a few ideas which may be useful for the paper.

EPIDEMIOLOGY OF RHINOSINUSITIS.  If space permits, I would probably say a little something about the epidemiology linking indoor mold and water exposure to sinusitis in either the introduction or discussion sections.  You might want to add something like this

“ Many published studies have linked indoor exposure to mold and water dampness to sinusitis.  A meta-analysis of 31 published studies reported that exposure to visual mold (OR 1.82, 95% CI 1.56-2.12) and indoor mold odor (OR 2.18, 95% CI 1.76-2.71) (Jaakkola et al. 2013).   A meta- analysis of 12 published studies reported that professional remediation of mold and water damaged homes was associated with significant reductions both in wheezing (OR 0.64, 95% CI -.55-0.75) and rhinitis (OR 0.57, 95% CI 0.49-0.66) (Sauni et al. 2015).

It seems to me that the epidemiological evidence seems to buttress the lab data linking fungal exposure to chronic rhinosinsitis.

PANT PAPER.   This paper discusses Aspergillus and Alternaria T Cell responses of allergic fungal rhinosinitus (Pant &Macardle 2014).  Maybe you want to cite this paper.

METHODOLOGY- FACS sorting and lab work.  Your methodology seems fine to me- but I have not done FACS work in several decades and perhaps some expert should double check your methodology. 

REFERENCES

Jaakkola MS, Quansah R, Hugg TT, Heikkinen SA, Jaakkola JJ (2013): Association of indoor dampness and molds with rhinitis risk: a systematic review and meta-analysis. J Allergy Clin Immunol 132, 1099-1110 e18

Pant H, Macardle P (2014): CD8(+) T cells implicated in the pathogenesis of allergic fungal rhinosinusitis. Allergy & rhinology (Providence, R.I.) 5, 146-56

Sauni R, Verbeek JH, Uitti J, Jauhiainen M, Kreiss K, Sigsgaard T (2015): Remediating buildings damaged by dampness and mould for preventing or reducing respiratory tract symptoms, infections and asthma. The Cochrane database of systematic reviews, Cd007897

Author Response

Manuscript Ickrath et al.: “Detection of Candida albicans-specific CD4+ and CD8+ T cells in blood and nasal mucosa of patients with chronic rhinosinusitis”

Manuscript ID: jof-1190818

Dear editor,

Thank you very much for giving us the possibility to revise the above mentioned MS in order to be publishable in Journal of Fungi. The referees’ comments proved to be very helpful to improve the MS and so we would like to submit a list of changes in response to the reviewers’ remarks. As we do receive an internal funding by the Open Access Publication Fund of the University of Wuerzburg if the manuscript will be accepted, I additionally added this information to Acknowledgments of the manuscript: "This publication was supported by the Open Access Publication Fund of the University of Wuerzburg."

Reviewer #3:

  • EPIDEMIOLOGY OF RHINOSINUSITIS. If space permits, I would probably say a little something about the epidemiology linking indoor mold and water exposure to sinusitis in either the introduction or discussion sections.  You might want to add something like this

“ Many published studies have linked indoor exposure to mold and water dampness to sinusitis.  A meta-analysis of 31 published studies reported that exposure to visual mold (OR 1.82, 95% CI 1.56-2.12) and indoor mold odor (OR 2.18, 95% CI 1.76-2.71) (Jaakkola et al. 2013).   A meta- analysis of 12 published studies reported that professional remediation of mold and water damaged homes was associated with significant reductions both in wheezing (OR 0.64, 95% CI -.55-0.75) and rhinitis (OR 0.57, 95% CI 0.49-0.66) (Sauni et al. 2015).

It seems to me that the epidemiological evidence seems to buttress the lab data linking fungal exposure to chronic rhinosinusitis.

Thank you for these two interesting studies. The following explanation was added to the discussion section.

Many published studies have linked indoor exposure to mold and water dampness to sinusitis. A meta-analysis of 31 published studies reported an increased risk for rhinitis and its subtypes in relation to mold odor, visible odor or exposure to dampness [27]. In addition to that, these findings were approved by a meta-analysis of 12 published studies that reported an improvement in wheezing and rhinitis through professional remediation of mold and water damaged homes [28]. These findings prove epidemiological evidence linking fungal exposure to chronic rhinosinusitis. (line 314-319)

  • PANT PAPER. This paper discusses Aspergillus and Alternaria T Cell responses of allergic fungal rhinosinitus (Pant &Macardle 2014).  Maybe you want to cite this paper.

Thank you for this advice, we added the citation of this manuscript to the main text. (Line 53)

Dear editor, if further data are needed to improve the MS, we will be happy to follow your suggestions. 

Yours sincerely,

Pascal Ickrath

Round 2

Reviewer 1 Report

The authors now addressed my major concerns. In my opinion it is now ready for publication

Reviewer 2 Report

The authors improved the manuscript by fixing the text, which became clearer for the reader.

The reason why I recommended the rejection of this manuscript (below) in the first round of review was not fixed. So the rejection of the present manuscript remains as my recommendation.

I understand that peripheral blood cells were used as controls for tissue-derived cells. However, a more important control would be to have tissue-derived cells (CRSsNP and CRSwNP) with (as in the graphs) and without (not in the graphs) the activating antigen. Without this information, it is impossible to state that any activation took place. It is interesting to compare with the activation status of peripheral blood cells, showing that tissue-derived cells are a polyclonal population that expanded in response to a previous contact with Candida`s antigens, but in my opinion, peripheral blood cells cannot be used to state that there was activation of tissue-derived cells, it would show that acivation is site-specific, but activation of T cells should be shown for any conclusion.  To state anything about activation of T cells, each condition should be shown in the presence (in the paper) and in the absence (not in the paper) of the activating antigen. As activation of T cells is the core of the study, this becomes critical for the interpretation of every result presented. The authors probably have the numbers for T cells (from every condition) in the absence of the activating peptide. Showing these results is crucial, otherwise other interpretations for the dataset become possible such as 
- There was no T cell activation in any experimental group analyzed, so there is no correlation between C. albicans and CRS in the evaluated conditions. 
- The basal expression of the activation marker Ki-67 is higher in T cells from tissues of patients with CRS than in peripheral blood cells. However, the incubation with Candida-derived antigen was not able to change the expression of the activation marker, so tissue-derived T cells from patients with CRS (all) are not responsive to Candida albicans or its antigens.

I am truly sorry to recommend the rejection of the manuscript, and hope that my comments can be useful to the authors.